# Health care professionals intention to use digital health data hub working in East Gojjam Hospitals, Northwest Ethiopia: Technology acceptance modeling

Ayenew Sisay Gebeyew[1]*, Sefefe Birhanu Tizie[1], Bayou Tilahun Assaye[1], Afework Edmealem[2], Temesgen Feyu[1], Habtamu Mekonen[3], Tirsit Ketsela Zeleke[4], Melese Getachew[4], Andualem Fentahun[1]

**1** Department of Health Informatics, College of Medicine and Health Science, Debre Markos University, Debre Markos, Ethiopia, **2** Department of Nursing, College of Medicine and Health Science, Debre Markos University, Debre Markos, Ethiopia, **3** Department of Nutrition, College of Medicine and Health Science, Debre Markos University, Debre Markos, Ethiopia, **4** Department of Pharmacy, College of Medicine and Health Science, Debre Markos University, Debre Markos, Ethiopia

* ayenew_sisay@dmu.edu.et

## Abstract

### Background

Digital health data hubs contribute significantly to finding the right solutions to health problems, which forms the basis for achieving sustainable development goals. However, in Ethiopia, the health system has been coming to one central hub for all data, there is limited evidence of health professionals' intentions to use these systems. Understanding their intentions is crucial, as this can significantly improve the advancement of digital health in healthcare organizations. This study assessed health professionals' intention to use digital health data hubs in hospitals in East Gojjam, northwest Ethiopia, in 2024.

### Methods

A cross-sectional study design was used to conduct the study. Eleven hospitals were included in the study area. Using an a priori structural equation modeling sample size calculator, the total sample size was 616. Stratified proportional allocation sampling was performed. The study participants were selected using a systematic sample. Structural equation modeling (SEM) was used for the analysis. Because it is a more powerful multivariate technique for testing and evaluating multivariate causal relationships. The assumptions of SEM-like normality, average variance extracted (AVE), composite reliability (CR), Cronbach's alpha, confirmatory factor analysis (CFA), and model specifications were checked using Amos and Stata version 16.

**Data availability statement:** All relevant data are within the paper and its Supporting Information files.

**Funding:** Funding was obtained from Research and Technology Transfer Directorate (RTTD) in Debre Markos University. This work would not be possible without the financial support of the Research and Technology Transfer Directorate (RTTD) under grant number 177/01/17. The funder had no role in study design, data collection and analysis, decision to publish, or preparation of the manuscript.

**Competing interests:** The authors have declared that no competing interests exist.

**Abbreviation:** AVE: Average Variance Extracted, CFA: Confirmatory Factor Analysis, CR: Composite Reliability, DH*2: Digital Health Data Hubs, ML: Maximum Likelihood, PCA: Principal Component Analysis, PEOU: Perceived Ease of Use, PU: Perceived Usefulness, SDG: Sustainable Development Goal, SEM: Structural Equation Modelling, TAM: Technology Acceptance Approach.

## Results

This study was conducted with a sample size of 616 healthcare professionals; 591 (95.94%) responded to the survey. The results showed that 57.69% (n = 341) of the healthcare professionals intended to use the digital health data hub. Further analysis showed that perceived usefulness (PU: β = 0.576, p = 0.000), perceived trust (PT: β = 0.116, p = 0.022), and attitude (β = 0.143, p = 0.043) significantly and positively influenced health professionals' intention to use digital health data hubs.

## Conclusion

Overall, the findings showed that 42.31% of health professionals have low intention to use digital health data hubs. These shall be needed to improve their intentions to use digital health data hubs through targeted interventions. Therefore, focusing on critical factors, such as perceived usefulness, trust, and attitude are crucial factors to reinforce their intention to use the system. Additionally, overcoming implementation challenges and building trust is critical to the successful integration and use of digital health data hubs.

## Introduction

### Background

Data are a central part of human life and have been stored in a standardized manner since the 18th century [1,2]. The global data space is remarkably large, continues to double every two years, and is expected to exceed 181 zettabytes by 2025 [2,3]. Although personal data storage is an offline source, cloud storage has recently become popular, for example, Apple iCloud, Dropbox, Google Drive, and Microsoft OneDrive [2,4,5]. In addition to this, new health technologies for healthcare services are emerging worldwide [6]. Therefore, digital healthcare users are taking advantage of developed storage technologies such as digital or cloud computing, which have made it possible to store enormous amounts of data efficiently [7].

Digital health solutions offer significant benefits and the potential to transform the healthcare system [8]. In addition, it will contribute to the implementation of the Sustainable Development Goals by 2030. All health-related information is stored and users retrieve relevant information for intervention purposes [8,9]. More people are gradually using mobile devices and the Internet for healthcare services. Digital health (DH) has significantly accelerated progress toward achieving the Sustainable Development Goals (SDGs) and strengthening healthcare systems in Africa [9,10]. According to a 2016 survey, approximately 24% of people used mobile health services, 20% used telemedicine, 17% used online learning, 6.7% used electronic health records, and 2.2% used big data [4,5].

Therefore, the establishment of digital health data hubs is the mainstay of providing the right solutions to health problems, as they act as a central location for finding, understanding, and implementing the fundamentals of quality primary healthcare

[11,12]. Ethiopia is currently at a developmental stage that allows the integration of data from various health services [13,14]. The Ethiopian Health Sector Transformation Plan shows that information transformation will be enhanced by strengthening digitized and cultivated health information through connected woreda, zones, and regions [15].

Therefore, healthcare professionals may have an additional responsibility to retrieve health information regarding patient history using digital health hubs [16]. Several electronic health systems are being implemented in Ethiopia to work with digital health data hubs. These systems are the Electronic Community Health Information System (eCHIS), electronic medical records (EMR), Healthnet, the District Health Information System (DHIS), Public Health Emergency Management (PHEM), and supply chain systems, which are in unified hubs [17]. Therefore, the use of connected health information or digital health data hubs (DH*2) is crucial to support wise health decisions and delivery of health services [18].

In the Ethiopian context, the health system has started using digital health devices and software to manage health care and health-related data [9,18]. However, it is still not fully utilized due to various obstacles such as the sparse distribution of pilot projects such as the Digital Health Hub, poor coordination, inadequate preparation of health workers for digital health, and lack of interoperability. Therefore, the data are displayed distantly, and looking up requires a long time. To adapt and modernize the healthcare system, the Ministry of Health introduced a digital healthcare system to unify healthcare data using a data hub to facilitate healthcare tasks [18]. However, there is insufficient evidence of health professionals' intentions to use digital health data hubs in resource-limited settings. For this reason, understanding the intentions of health professionals makes an enormous contribution to digital health stakeholders, especially health professionals who play a leading role in the field of health care [19]. Investigating digital health data hubs offers a significant opportunity to improve health systems and achieve health-related SDG goals. This study contributes to building a strong healthcare system to help healthcare professionals, especially those working remotely, access fair and high-quality healthcare data that replaces data disintegration.

## Theoretical models

There are two sections in the literature about digital health data hubs. The first part discussed digital health data hubs and technology adoption models. In the second part, details of the hypothesis for the development of digital health data hubs and the conceptual framework of the study were presented.

**Digital health data hub and technology acceptance model.** Stored health data is of utmost importance to organizations and individuals when making decisions. Nowadays, thanks to the Internet, online digital storage services are replacing offline data storage [17,20]. It is important to examine emerging cloud-based adoption at the organizational level. Healthcare organizations have continually recorded data over time for customers, suppliers, and stakeholders to analyze this information and derive insights. Research on the extent of digital storage adoption in developing countries is still in its early stages, and few studies have focused on the factors that influence individual adoption [20]. Researchers have limitations when it comes to digital health hubs, even though the technology is growing rapidly. To identify the gap, an expanded version of the Technology Acceptance Model was developed and tested to measure healthcare professionals' behavioral intentions.

Numerous models have been developed to estimate the intent and acceptance of technology and information systems. However, the Technology Acceptance Model (TAM) is the most commonly used approach [7]. In 1987, Fred Davis created a model that illustrates the relationship between variables and behavioral intentions to adopt a particular technology. According to the paradigm, behavioral intention (BI) is influenced by attitude (ATT), which in turn is influenced by perceived usefulness (PU) and ease of use (PEOU) [21]. The results of studies on behavioral intention when using online-based digital systems support this model [20]. Studies have shown that perceived usefulness and ease of use positively influence attitudes, which in turn influence users' intentions to use digital systems [22,23].

**Hypothesis development.** The main focus here is the user's perception of the extent to which the use of a particular technology improves the user's performance [7]. It showed that users were looking for a cloud-based digital system

to store their digital health data. In addition to the Technology Acceptance Model (TAM), studies show that perceived usefulness influences users' attitudes and intentions toward using digital health [24].

Based on TAM theory, a system may be considered more useful if the user perceives that it is easy to use [21]. Perceived ease of use depends on the extent to which a person believes that using a digital health data hub is not time-consuming and requires a lot of effort. Therefore, studies support that perceived ease of use has a significant impact on perceived usefulness, attitude, and intention to use digital cloud services [23]. In addition to the proposed model of technology acceptance, studies have shown that health professionals' attitudes have a strong influence on their intention to use eHealth systems [24]. In terms of data storage, the setting has an impact on the use of digital health data hubs [7]. A study shows that attitudes towards digital data storage have a positive behavioral intention when using the systems [22].

In addition perceived trust is a crucial parameter when exchanging information using digital systems, especially when transmitting sensitive personal and institutional data and information [25]. Individuals have counteracted the potentially negative influence of privacy concerns associated with the adoption of digital health data hubs [26]. Studies have shown that trust is a significant factor in perceived usefulness, usability, and user intentions [23,27].

In addition, the practice of information sharing is the core task of health professionals, which influences the behavior of health professionals in terms of the intention to use digital health data [28,29]. Therefore, information flow within cloud-based healthcare platforms results in efficient resource utilization for service providers and users [30]. One study shows that access to shared information increases behavioral intention to use digital health data hubs [31].

However, the perceived risks mainly come from the data in the cloud, which can be affected by the loss of private information or the user's assessment of the severity of the adverse consequences of digital health systems [7]. Although the emerging digital health technology brings with it a lot of convenience, there is a perceived risk as the process is carried out over the Internet. Therefore, private information can be illegally stolen and accessed by an authorized entity [32]. One study shows that perceived risk negatively influenced adoption intention; however, it is less risky to see a benefit (25). Similarly, a study shows that perceived risk has an impact on reducing perceived trust and attitude towards digital health data hubs [23,33].

Based on this literature, the following hypothesis was formulated

H1: Perceived ease of use positively influenced attitudes toward the digital health data hub.

H2: Perceived trust positively influenced the perceived usefulness of the digital health data hub.

H3: Perceived risk negatively influenced the perceived usefulness of the digital health data hub.

H4: Perceived trust positively influenced attitudes toward the digital health data hub

H5: Perceived risk negatively influenced attitudes toward the digital health data hub.

H6: Perceived ease of use positively influenced the perceived usefulness of digital health data hub

H7: Perceived ease of use positively influenced behavioral intention of digital health data hub.

H8: Attitude toward using a digital health data hub positively influenced usage intention.

H9: The perceived usefulness of the digital health data hub positively influences behavioral intention.

H10: Perceived trust in the digital health data hub positively influences behavioral intention to use.

H11: Perceived risk in the digital health data hub negatively influences behavioral intention.

H12: Perceived usefulness of the digital health data hub positively influences attitude.

H13: Perceived risk in the digital health data hub has a negative impact on perceived trust.

H14: The practice of information sharing positively influenced behavioral intention

In summary, the relationship evolved from TAM-constructed variables and additional variables from previous studies. Perceived usefulness, perceived ease of use, attitude, and behavioral intention to use were taken from the technology acceptance model. Perceived trust, perceived risk, and information-sharing practice were added according to the kinds of literature (Fig 1). Therefore, to test the research questions how the determinants influenced health professionals' intention to use digital health data hubs is discussed in the next method section.

## Method and materials

### Study design and period

A cross-sectional study was conducted to assess the intention to use digital health data hubs among healthcare professionals at East Gojjam Hospital, in northwest Ethiopia. The study was conducted from May 30 to August 30, 2024, in East Gojjam hospitals.

### Study area

The study was conducted in East Gojjam, Amhara Region, Ethiopia. There are eleven hospitals in this area, where around 2,551 health professionals work. The study site was in the town of Debre Markos in the east of Gojjam, 255 km, and 296

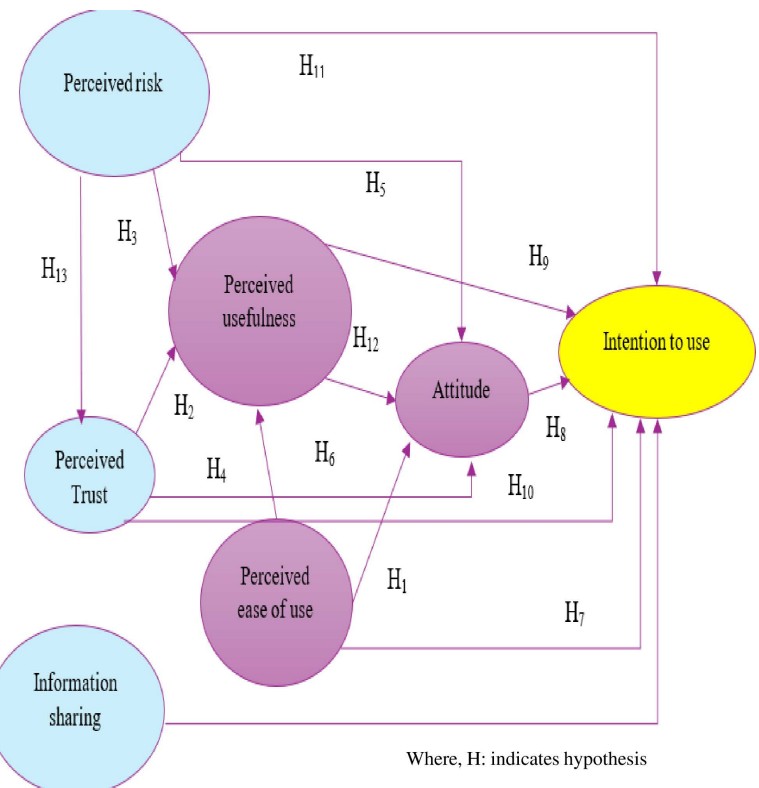

Where, H: indicates hypothesis

**Fig 1. Conceptual framework TAM to use digital health data hub.**

km from Bahari Dar and Addis Ababa, respectively. In this area, there is one specialized referral hospital, one general hospital, and nine primary hospitals. These are serving the community by providing laboratory, pharmacy, radiology, intensive patient care, ophthalmic, psychiatry, obstetrics, and pediatric services for the community. In these hospitals, health professionals including medical doctors, pharmacy, Anastasia, optometry, laboratory, radiology, health informatics, nurses, midwifery, psychiatry, and health officers were included in the study.

### Study population and sample size determination

All health professionals working in East Gojjam Hospital in northwestern Ethiopia were a source population, and health professionals using digital devices in East Gojjam Hospital were selected for the study population during the data collection period. Health professionals who used digital health data were then included in the study. Structural equation modeling requires a large sample technique to obtain sufficient power for individual parameter tests such as factor loadings [34,35]. To calculate sample size, an a priori structural equation modeling sample size calculator was used to calculate the required sample size for a study using a structural equation model (SEM) with 28 observed variables, 7 latent variables in the model, and an expected value of 0.1758 Effect size, 0.05 desired probability and 0.8 statistical power levels [36]. Therefore, the recommended minimum sample size is 560. With a non-response rate of 10%, the sample size was 616.

### Sampling procedure

There are eleven hospitals in the East Gojjam Zone in northwest Ethiopia with a total of 2,551 Health professionals. A proportional allocation was made for each health facility and staff unit. Then, first, stratified sampling was used to create a stratum at the health facility level, as there are specialized, general, and primary hospitals. The study participants were then selected using systematic sampling from a list of human resources records (Fig 2).

### Study variable and operational definition

The study defined a model with two types of construct variables. These are constructs that are used directly from the TAM model. For testing purposes, three additional constructs were added to the TAM: perceived risk, perceived trust, and information sharing. These constructs were uncovered at the time of the literature review. According to the literature, five types of Likert scale questions were developed to measure the parameters. Digital Health Data Hub: The epicenter of health data, organized into one entity for easy access. Intention to use the digital health data hub: - The degree to which health professionals have intended to use the digital health data hub or not. This construct was measured using four-point questions and achieved a score of over 50%. They have the intention to use the digital health data hub, whereas less than 50% have no intention to use the digital health data hubs [7,20,37,38].

   Attitude towards the digital health data hub: - Health professionals' feeling is good or not to use the digital health data hub, measured by three item questions [7,37].

   Perception of usefulness: - The degree to which healthcare professionals believe that using the digital health data hub contributes to improved work performance. This construct is measured with eight-item questions [7].

   Perceived ease of use: - The level of ease of use when dealing with digital health data, measured using four-item questions [7,37].

   Perceived risk: - Healthcare professionals' individual willingness to use the digital health data hub may or may not be risky and is measured using three items [7,23].

   Perceived trust: - Healthcare professionals' perception that the digital health data hub may need to respect the digital policy, measured using four items [7].

## Data collection procedure and quality assurance

The target population of the main study was healthcare professionals working at East Gojjam Hospital in northwest Ethiopia. An interview questionnaire was administered (S1 Appendix). Before that, this study adapts a survey research design from TAM constructs and slightly adapts the items to fit the empirical context of the study [7]. To reliable and valid the instrument, the first task was to assign a research team from different health departments who have profound knowledge and were trained in digital health to review and identify discrepancies regarding the research topic. Then, the questionnaire was prepared, and a research consultation was made. Based on suggestions from medical experts including health professionals and research consultants, the queries were worded precisely and carefully. Furthermore, a pre-test was conducted to check the performance of the representative survey in selected hospitals in West Gojjam. Data were collected from May 30 to August 30, 2024 by eight data collectors under the supervision of eight supervisors. The data

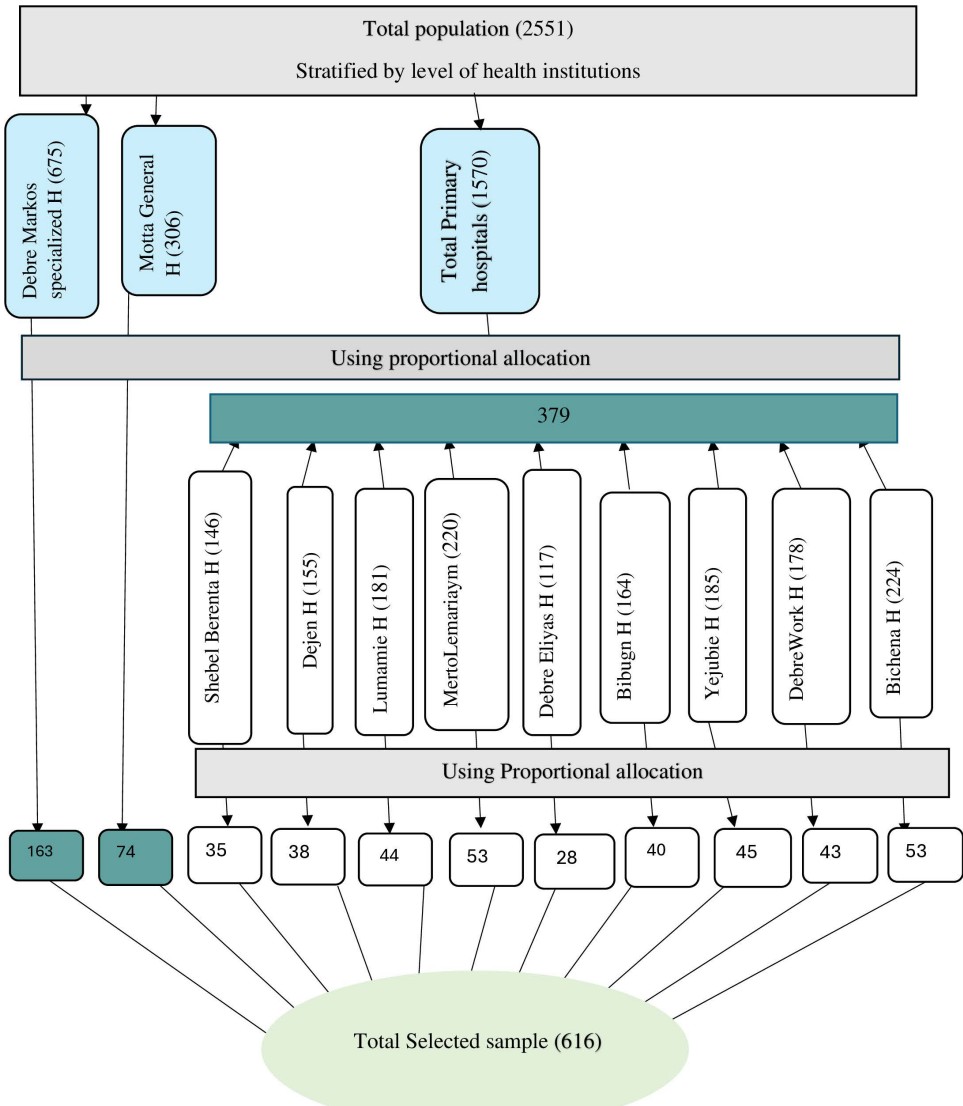

**Fig 2. Sampling procedure of health professionals in East Gojjam hospitals, Ethiopia, 2024.**

was also collected by health informatics professionals and training was conducted to implement the research objective. In detail, the study used scales with four items for perceived ease of use, three items for behavioral intention, eight for perceived usefulness, three items for perceived risk, four items for perceived trust, and three items for attitude toward use [7,33]. The items for all constructs were measured using five-point Likert scales, with a range from "strongly disagree" (1) to "strongly agree" (5).

Some screening questions were asked to indicate exposure related to digital data storage and background questions were also included for descriptive analysis. After administered actual survey, principal component analysis (PCA) was used to validate each item. Missing values were checked, and the assessment of univariate and multivariate normality was checked using skewness and kurtosis values. Factor loadings were examined and values above 70 were included in the final measurement model. Average variance extracted (AVE) and correlation were also assessed for all constructs. In addition, internal consistency was assessed by calculating composite reliability (CR) and Cronbach's alpha. Confirmatory factor analysis (CFA) was used to examine validity and reliability using Amos and Stata version 16.

### Data management and analysis

Structural equation modeling (SEM) with maximum likelihood estimation (ML) was applied using AMOS software. The measurement model is verified. In the cross-sectional survey, the data were used for estimation to test the hypothesized relationships and possible influences between the model constructs. After defining the model criteria, the sample size was sufficient to carry out the structural model.

### Ethical considerations

A study was conducted after obtaining ethics clearance approval from the Ethics Review Board of the Review Board (IRB) of the College of Health Science, Debre Markos University (No.325/01/16). Approval letters were then obtained from Amhara Regional Health Breau and distributed to each hospital. Ultimately, written informed consent was obtained from each research participant.

## Results

### Sociodemographic characteristics

A total of 616 healthcare professionals were involved in this research. Overall, 95.94 percent (n = 591) of healthcare professionals completed the study. The mean age of participants was 32.198 ± 4.89 SD and most were between 31 and 40 years old (50 percent, n = 297). Based on the gender distribution of the participants, 49.92 percent (n = 295) were female and 50.08% (n = 296) were male. Of the healthcare professionals, 63 percent (n = 375) were married, 25 percent (n = 149) were single, and 11 percent (n = 67) were other. In the specialty, nurses (23.86 percent, n = 141), midwives (16.24 percent, n = 96), and pharmacists (14.55 percent, n = 86) had the highest proportion of health professionals (Table 1).

### Health professional characteristics towards digital health data hubs

According to the study results, the majority of study participants, namely 63.11 percent (n = 373), were convinced of the benefits of health data hubs. Most healthcare professionals also have a good attitude towards the health data hub, which accounts for 68.02 percent (n = 402). In terms of perceived trust, half of the study participants found digital health data hubs to be trustworthy, which is 49.75 percent (n = 294). When it comes to digital risk perception, 46.36 percent (n = 264) of healthcare professionals have concerns about healthcare data hubs. Additionally, the key finding of healthcare professionals' intention to use digital health data hubs was 57.69 percent coverage (n = 341), as shown (Table 2).

**Table 1. Sociodemographic characteristics of health professionals in East Gojjam hospitals in Northwest Ethiopia, 2024.**

| Age | Freq. | Percent |
|---|---|---|
| Less than 30 years | 264 | 44.67 |
| 31-40 years | 297 | 50.25 |
| Above 40 years | 30 | 5.08 |
| **Gender** | | |
| Female | 295 | 49.92 |
| Male | 296 | 50.08 |
| **Marital status** | | |
| Single | 149 | 25.21 |
| Married | 375 | 63.45 |
| Others (Separated, Divorced) | 67 | 11.33 |
| **Health Profession** | | |
| Anastasia | 32 | 5.41 |
| Health informatics | 32 | 5.41 |
| Health officer | 26 | 4.40 |
| Medical laboratory | 55 | 9.31 |
| Midwives | 96 | 16.24 |
| Nurse | 141 | 23.86 |
| Pharmacy | 86 | 14.55 |
| Physician | 63 | 10.66 |
| Radiologist | 24 | 4.06 |
| Other (Optometry, Psychiatry& Physiotherapy) | 35 | 6.08 |
| **Educational status** | | |
| Diploma | 38 | 6.43 |
| Bachelor degree | 361 | 61.08 |
| Master's degree | 129 | 21.83 |
| Doctors | 62 | 10.65 |

## Measurement model

AMOS was used to evaluate the proposed model. After checking the criteria of the measurement model, structural models were also evaluated. Therefore, it is helpful to evaluate models with many relationships, indicators, and constructs very well. AMOS version 4.0.8.4 was used to test the proposed model. The first step was to test the reliability and validity of the measurement model (Fig 3). The measurement model was then evaluated using the following parameters. These are item loadings, composite reliability, Cronbach's alpha, convergent, and discriminant validity (Table 3). Construct reliability indicates how consistently the variables reflect the intended concept [39]. Given the diversity of measurements used, it was critical to use Cronbach's alpha and composite reliability to verify that each construct maintained a unique and nonoverlapping identity. In addition, the uniqueness of constructs was examined using discriminant validity [40].

Convergent validity based on average variance extracted (AVE) confirms that the indicators intended to represent the same idea successfully fit the representation of the underlying construct. It was measured with the AVE. This helps to show the percentage of variance in the indicators explained by the latent construct and the AVE value is greater than 0.50. This is a threshold for the successful presentation of the indicators [39]. Cronbach's alpha (CA), element loadings, composite reliability scores (CR), and design AVE are all listed in Table 3. In AMOS-SEM, composite reliability is a key metric. Scholars suggested, values between 0.60 and 0.70 are considered acceptable, while higher values above 0.70 are considered reliable. The internal consistency of the model was checked. The result shows that all Cronbach's alpha and

**Table 2. Health professional's characteristics of Digital Health Data Hub in East Gojjam Hospitals in Northwest Ethiopia, 2024.**

| Perceived usefulness | | Frequency | | Percent | |
|---|---|---|---|---|---|
| Inadequate | | 218 | | 36.89 | |
| Adequate | | 373 | | 63.11 | |
| **Perceived ease of use** | | | | | |
| Not easy | | 243 | | 41.12 | |
| Ease | | 348 | | 58.88 | |
| **Attitude** | | | | | |
| unfavorable | | 189 | | 31.98 | |
| Favorable | | 402 | | 68.02 | |
| **Perceived trust** | | | | | |
| Not trusted | | 297 | | 50.25 | |
| Trusted | | 294 | | 49.75 | |
| **Perceived risk** | | | | | |
| Not perceived risk | | 317 | | 53.64 | |
| Perceived risk | | 274 | | 46.36 | |
| **Behavioral intention** | Frequency (%) | Proportion | Std. Err. | | 95% Conf. Interval |
| Poor intention | 250(42.30) | .4230 | .0203 | | 0.384- 0.463 |
| Have intention | 341(57.69) | .5769 | .0203 | | 0.537 - 0.616 |

composite reliability values were above the cutoff value of 0.60 and 0.70, respectively [41]. Each AVE result exceeded the recommended minimum value of 0.50 for each relevant component in terms of convergent validity [41]. The constructs demonstrated discriminant validity that exceeded the acceptable level based on the squared average variance extracted values. Therefore, the squared average variance extracted values outperform the correlation construct values as shown in a table (Table 4). The model also fits well as shown in the table (Table 5).

## Findings of the study

To examine how the constructs relate to each other, a structural model of the research model was created. Bootstrapping methods were used to evaluate the model using 500 iterations. Then the path coefficient and a 95 percent confidence interval were used. The overall fit statistics of the structural model showed strong fit: $\chi^2/df = 2.260$, SMRS = 0.0316, RMSEA = 0.043, CFI = 0.975, AGFI = 0.926 and GFI = 0.946. The results of the structural model sections clearly show that the model proposed in this study fits the data quite well (Fig 4). The model shows H1, H4, H5, H6, H8, H9, H10 and H12 which support the hypothesis. On the other hand, H2, H3, H7, and H11 do not support the hypothesis (Table 6).

After testing the hypothesis, perceived usefulness, trust, and attitude of healthcare professionals significantly influenced behavioral intention to use digital health data hubs. The model revealed that 59% ($R^2 = 0.59$) of the variance in health professionals' intention to use digital health data hubs was influenced by these factors. Table 6 shows the results of the study in which the hypothesis was evaluated using path coefficients ($\beta$) and t-statistics. According to the results, perceived ease of use (H1: $\beta = 0.261$, p = 0.004), trust (H4: $\beta = 0.121$, p = 0.039) and usefulness (H12: $\beta = 0.376$, p = 0.001) were perceived. They have a positive impact on health professionals' attitudes towards health data hubs. In contrast, perceived risk does not have a significant impact on attitude toward the health data hub ($\beta = 0.098$, p = 0.073), leading to the rejection of H5. The result shows that perceived ease of use has a significant influence on perceived usefulness ($\beta = 0.647$, p = 0.000). Therefore, hypothesis H6 was confirmed. Finally, attitude ($\beta = 0.143$, p = 0.043), usefulness ($\beta = 0.576$, p = 0.000), and perceived trust ($\beta = 0.116$, p = 0.022) were identified as significant factors influencing health professionals' intention.

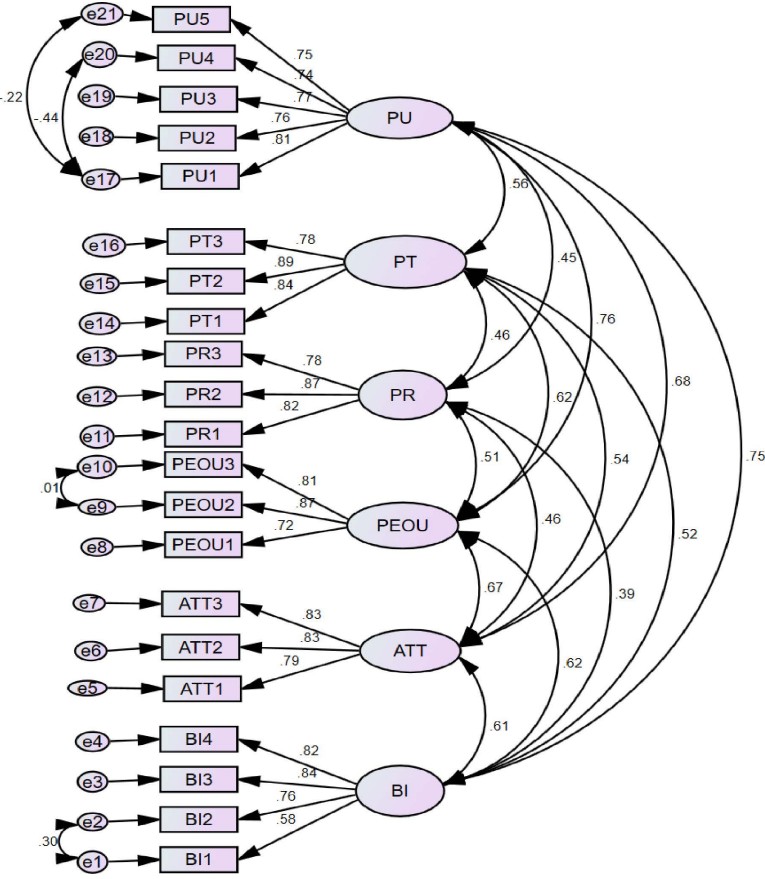

**Fig 3. Measurement model assessment of intention to digital health data hub in East Gojjam hospitals in Northwest Ethiopia 2024.** PR: perceived risk, PT: perceived trust, PEOU: Perceived ease of use, PU: Perceived usefulness, ATT: Attitude, BI: Behavioral Intention, e: error.

## Mediation of the study

The estimator was used to examine models by determining the relationships of independent variables on the effects of outcome variables. This helps determine how the explanatory variable indirectly influences the dependent variable through a variety of mediators [42].

Attitude and perceived usefulness played a role as mediators in the relationship between perceived ease of use and behavioral intention. Health professionals' behavioral intention was significantly influenced by perceived ease of use through attitude (β = 0.078, t = 2.053, p = 0.022), supporting the hypothesis. Similarly, perceived ease of use significantly influenced behavioral intention through perceived usefulness (β = 0.329, t = 3.323, p = 0.000), supporting this hypothesis. This study also examined the relationships between perceived risk, perceived trust, and behavioral intention, as well as the mediating roles of attitude and perceived usefulness.

According to the results, attitude toward the digital health data hub had a non-significant indirect influence on behavioral intention through perceived trust (β = 0.012, t = 1.20, p = 0.170), contradicting the hypothesis. Perceived risk (β = 0.009, t = 1.125, p = 0.237) has no significant indirect influence on behavioral intention through attitude rejecting support for the hypothesis. The hypothesis also rejects the idea that perceived risk influences behavioral intention indirectly through perceived benefit (β = 0.008, t = 0.363, p = 0.741) (Table 7).

## Discussion

**Table 3. Present the model validation results at each of the points of measurement.**

|  | Item | Loadings | Composite Reliability (CR) | Cronbach's alpha (Cronbach's α) | AVE |
|---|---|---|---|---|---|
| BI | BI1 | 0.577 |  |  |  |
|  | BI2 | 0.762 |  |  |  |
|  | BI3 | 0.843 |  |  |  |
|  | BI4 | 0.820 | 0.841 | 0.845 | 0.574 |
| ATT | ATT1 | 0.791 |  |  |  |
|  | ATT2 | 0.828 |  |  |  |
|  | ATT3 | 0.832 | 0.859 | 0.8584 | 0.669 |
| PEOU | PEOU1 | 0.721 |  |  |  |
|  | PEOU2 | 0.866 |  |  |  |
|  | PEOU3 | 0.806 | 0.841 | 0.8369 | 0.639 |
| PR | PR1 | 0.823 |  |  |  |
|  | PR2 | 0.872 |  |  |  |
|  | PR3 | 0.785 | 0.867 | 0.8648 | 0.685 |
| PT | PT1 | 0.839 |  |  |  |
|  | PT2 | 0.887 |  |  |  |
|  | PT3 | 0.780 | 0.875 | 0.8710 | 0.700 |
| PU | PU1 | 0.809 |  |  |  |
|  | PU2 | 0.764 |  |  |  |
|  | PU3 | 0.765 |  |  |  |
|  | PU4 | 0.737 |  |  |  |
|  | PU5 | 0.754 | 0.877 | 0.8638 | 0.587 |

**Table 4. Construct discriminate validity.**

| Construct | BI | ATT | PEOU | PR | PT | PU |
|---|---|---|---|---|---|---|
| BI | **0.758** |  |  |  |  |  |
| ATT | 0.610*** | **0.818** |  |  |  |  |
| PEOU | 0.618*** | 0.671*** | **0.800** |  |  |  |
| PR | 0.388*** | 0.456*** | 0.510*** | **0.827** |  |  |
| PT | 0.524*** | 0.538*** | 0.619*** | 0.464*** | **0.837** |  |
| PU | 0.750*** | 0.685*** | 0.757*** | 0.447*** | 0.558*** | **0.766** |

**Table 5. Model fit summary of the study.**

| Matrix Fit indices | Threshold value (cut-off point) | Results of the study | Interpretation |
|---|---|---|---|
| Chi-Square/Degree of Freedom | < 3 | 2.087 | Accepted |
| The Goodness of Fit Index (GFI) | > 0.9 | 0.946 | Accepted |
| Adjusted Goodness of fit index (AFGI) | > 0.8 | 0.926 | Accepted |
| Comparative Fit Index (CFI) | > 0.9 | 0.975 | Accepted |
| Tucker-Lewis's index (TLI) | > 0.9 | 0.969 | Accepted |
| Root means square error of approximation (RMSEA) | < 0.08 | 0.043 | Accepted |
| Standardized root means square residual (SRMR) | < 0.08 | 0.0316 | Accepted |

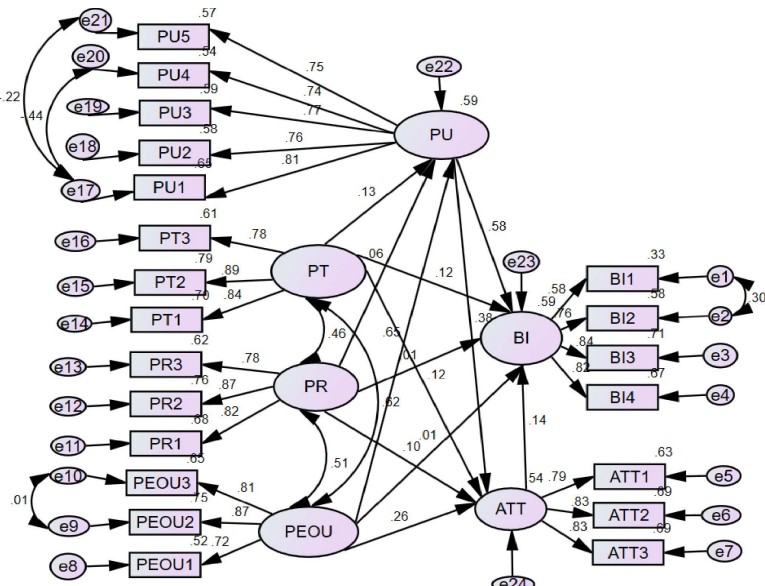

**Fig 4. Structural model assessment of intention to use digital health data hub among health professionals working at East Gojjam hospitals in northwest Ethiopia 2024.**

**Table 6. Structural model assessment and Hypothesis test results (n = 591).**

| Hypothesis | Construct | Estimate(β) | S. E. | t-statistics | 95% of CI | P | Decision |
|---|---|---|---|---|---|---|---|
| H1 | PEOU ->ATT | 0. 261 | 0.092 | 3.584 | 0. 084- 0. 446 | 0.004 | Accepted |
| H2 | PT->PU | 0.131 | 0.064 | 2.666 | 0.005–0.252 | 0.040 | Accepted |
| H3 | PR->PU | 0.055 | 0.049 | 1.299 | -0.041 - 0.151 | 0.261 | Not accepted |
| H4 | PT->ATT | 0. 121 | 0.058 | 2.420 | 0.007 - 0.235 | 0.039 | Accepted |
| H5 | PR->ATT | 0.098 | 0.056 | 2.218 | -0.010 - 0.210 | 0.073 | Not Accepted |
| H6 | PEOU->PU | 0.647 | 0.071 | 10.539 | 0.506 - 0.787 | *** | Accepted |
| H7 | PEOU->BI | 0. 011 | 0.101 | 0.160 | -0.181 - 0.204 | 0.895 | Not accepted |
| H8 | ATT->BI | 0.143 | 0.071 | 2.480 | 0.005 - 0.282 | 0.043 | Accepted |
| H9 | PU->BI | 0. 576 | 0.098 | 7.509 | 0.381 - 0.769 | *** | Accepted |
| H10 | PT->BI | 0. 116 | 0.051 | 2.349 | 0.016 - 0.219 | 0.022 | Accepted |
| H11 | PR->BI | 0. 006 | 0.053 | 0.131 | -0.080 - 0.088 | 0.895 | Not Accepted |
| H12 | PU->ATT | 0. 376 | 0.083 | 5.788 | 0. 205- 0.532 | 0.001 | Accepted |

Variables with

***indicate statistically significant at p-value < 0.001 PR: Perceived Risk, PT: Perceived Trust, PEOU: Perceived ease of use, PU: Perceived usefulness, ATT: Attitude, BI: Behavioral Intention.

The purpose of this study was to determine whether healthcare professionals intend to use technology-enabled health data hubs. Out of the total, 57 percent of study participants (95 percent CI: 0.537–0.616) intend to use the digital health data hubs. This study shows that there is a moderate to high intent to use digital health data hubs in healthcare organizations with limited resources. This showed that health professionals have limited intention towards digital health data hubs. This is due to the limited interaction of health professionals with digital health applications. Compared with a study conducted in South Australia, 98% of the participants have an intention to use a digital health data hub [16]. This is due

**Table 7. Mediation analysis of the model.**

| Hypothesis | Construct | Estimate (β) | SE | t-statistics | Effect | P-value | Types of mediation |
|---|---|---|---|---|---|---|---|
| H13 | PEOU->ATT->BI | 0.078 | 0.038 | 2.053 | Indirect<br>Direct<br>Total | 0.022<br>0.953<br>0.000 | Complete mediation |
| H14 | PEOU->PU->BI | 0.329 | 0.099 | 3.323 | Indirect<br>Direct<br>Total | 0.000<br>0.953<br>0.000 | Complete mediation |
| H15 | PT->ATT->BI | 0.012 | 0.010 | 1.20 | Indirect<br>Direct<br>Total | 0.170<br>0.031<br>0.028 | Insignificant |
| H16 | PR->PU->BI | 0.008 | 0.022 | 0.363 | Indirect<br>Direct<br>Total | 0.741<br>0.927<br>0.613 | Insignificant |
| H17 | PR->ATT->BI | 0.009 | 0.008 | 1.125 | Indirect<br>Direct<br>Total | 0.237<br>0.927<br>0.613 | Insignificant |

to caregivers' thoughts digital health hubs could improve health outcomes. Moreover, a systematic review involved health professionals including nurses, doctors, and mixed samples of physicians, signifying that all healthcare profiles have been involved to date, although to a limited extent for some health professionals [43]. Therefore, improving the intent rates of digital health data hubs is essential for the delivery of healthcare services [44,45].

The study identified health professionals' attitudes and intentions toward using the digital health data hub. Perceived usefulness (β = 0.376, p = 0.001), perceived trust (β = 0.121, p = 0.039) and perceived ease of use (β = 0.261, p = 0.004) are the key factors associated with healthcare professionals' attitude towards digital health data hubs, as shown in Table 6. This finding is consistent with other research showing how perceived usefulness, trustworthiness, and ease of use influence attitudes toward cloud services in healthcare [23]. Likewise, physicians' perceptions of trust influence how they think about their intention to use digital health data hubs [46]. Furthermore, a systematic review showed that the digital native generation has more attitudes toward digitalization [43], and this suggests that improving the perceptions regarding usefulness, ease of use, and trust in digitation might be relevant to mature healthcare professionals.

As the result showed that, three main factors influence health professionals' intention to use health data hubs. According to this result, first, perceived usefulness (β = 0.576, p = 0.000) significantly influenced health professionals' intention to use digital health data hubs. Hence, the intention to use digital health data hubs is still mainly focused on health care professionals' perspectives [43]. This finding is consistent with a study conducted in America that perceived usefulness was positively associated with intent to use digital health applications [47]. Another piece of evidence supports this finding and shows that users wanted to use digital data hubs because they benefited greatly from the digital health platform. Therefore, the perceived usefulness of digital health data hubs influences their intentions to use the system [24].

Second, health professionals' attitudes had a significant influence on the intention to use health data hubs (β = 0.143, p = 0.043). Hence, improving the attitude of health professionals has the ultimate purpose of enhancing their intentions towards digital data hubs. This is supported by evidence that shows the positive attitude of healthcare professionals is a key factor in their propensity to use digital health data hubs [24]. However, in this study, 68.02%(n = 402) of study participants have a favorable attitude towards digital health data hubs. Thus, the level of attitude should be enhanced by improving perceived usefulness, perceived ease of use, and trust as shown in Table 6.

The third determining factor influencing health professionals' intention to use digital health data hubs was perceived trust (β = 0.116, p = 0.022). The main problem in the provision of digital health services is therefore the perceived trustworthiness of digitally enabled health applications. Hence, improving their perceived trust enables higher intentions to

use digital health data hubs. Evidence has shown that trust is critical when dealing with personal or institutional health information over digital networks [25]. Additionally, other studies showed that perceived trust has a great impact on users' intentions toward digital health apps [23,27].

Finally, the study revealed the significant indirect effects of health professionals' intention affecting perceived usefulness ($\beta = 0.329$, $p = 0.000$) and perceived ease of use through attitude ($\beta = 0.078$, $p = 0.022$). Hence, positively perceived usefulness and ease of use have a high degree of effect on health professionals' intention to use digital health data hubs indirectly. The finding is supported by the interdependence of the Technology Acceptance Model (TAM) constructs and evidence revealed that the intention to use digital health services is highly dependent on perceived ease of use and perceived usefulness both directly and indirectly [23,48].

### Strength and limitations

This study examines the creation of a health network through a digital health data hub and provides important insights into the development of digital health systems and security. It uses structural equation modeling according to the Technology Acceptance Model. However, the study's cross-sectional design may limit its ability to identify cause-and-effect relationships.

### Conclusion

This study found that health professionals' attitudes, perceived usefulness, and perceived trust have a significant impact on their intentions to use digital health data hubs. These findings are essential to develop integrated digital health hubs which are trusted and accessible digital health solutions. This helps to effectively increase the intention levels of health professionals to use digital health data hubs. To encourage wider adoption, higher priority shall be given to resolving usability issues and building trust. By focusing on these elements, stakeholders can ensure the long-term viability of digital health technologies in the healthcare environment and promote their effective use. In today's world, integrated health care is provided holistically by aided digital health data hubs. Hence, in the future, investigating digital health data hubs linked to health professionals' practical content will be the most relevant for networked healthcare settings.

### Implications

The study suggests that the intention of health professionals toward digital health data hubs was not adequate. It showed that more than 42 percent of health professionals have low intention to use digital health data hubs. The research implies an increase in the intention of health professionals towards digital health data hub aids for managing routine health care activities. This enhancement could be through advancing their perception regarding digital health hubs' usefulness, digital trust, and attitudes. This helps health professionals can be rich in qualified health information for making decisions. Also hospital administrators, policymakers, and others can retrieve tangible information for decision-making by connecting woreda, regional and tertiary level hospitals, or using digital health data hubs.

### Recommendation

The results will be forwarded to the Federal Ministry of Health, Amhara Health Bureau, Debre Markos University, East Gojjam Referral Hospitals, and other relevant stakeholders. To promote digital trust, which in turn promotes the use of digital tools, the goal was to increase healthcare professionals' intentions to use digital health data platforms. These results can also be used as a starting point for further studies and help advance digital health in healthcare.

### Supporting information

**S1 Appendix. Questionnaires of the study.**
(DOCX)

**S2 Data. A portion of data about health care professionals intention to use digital health data hub.**
(DTA)

## Acknowledgments

First and foremost, we extend our deepest gratitude to the compassionate God. We also sincerely appreciate the Research and Community Service Vice President's Office of Debre Markos University for their commitment to encouraging staff to identify societal problems and address them using scientific methods. Then, we are deeply thankful to the College of Medicine and Health Sciences at Debre Markos University for their efforts in fostering research initiatives within the healthcare setting. Finally, we express our heartfelt appreciation to the Health Informatics staff and our colleagues for their unwavering support in the development of these research papers.

## Author contributions

**Conceptualization:** Ayenew Sisay Gebeyew, Bayou Tilahun Assaye, Afework Edmealem.

**Data curation:** Ayenew Sisay Gebeyew, Tirsit Ketsela Zeleke.

**Formal analysis:** Ayenew Sisay Gebeyew.

**Funding acquisition:** Ayenew Sisay Gebeyew, Bayou Tilahun Assaye, Afework Edmealem, Habtamu Mekonen, Tirsit Ketsela Zeleke, Melese Getachew, Andualem Fentahun Senishaw.

**Investigation:** Ayenew Sisay Gebeyew, Sefefe Birhanu Tizie.

**Methodology:** Ayenew Sisay Gebeyew, Habtamu Mekonen, Melese Getachew.

**Project administration:** Ayenew Sisay Gebeyew.

**Resources:** Ayenew Sisay Gebeyew, Sefefe Birhanu Tizie.

**Software:** Ayenew Sisay Gebeyew, Sefefe Birhanu Tizie.

**Supervision:** Ayenew Sisay Gebeyew, Sefefe Birhanu Tizie.

**Validation:** Ayenew Sisay Gebeyew, Sefefe Birhanu Tizie.

**Visualization:** Ayenew Sisay Gebeyew, Temesgen Feyu.

**Writing – review & editing:** Ayenew Sisay Gebeyew, Sefefe Birhanu Tizie, Bayou Tilahun Assaye, Afework Edmealem, Habtamu Mekonen, Tirsit Ketsela Zeleke, Melese Getachew, Andualem Fentahun Senishaw.

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
