## [Decision Letter · Decision Letter 0]

24 Feb 2025

PONE-D-25-02270Health Care Professional’s Intention to Use Digital Health Data Hub Working in East Gojjam Hospitals, Northwest Ethiopia: Technology Acceptance ModelingPLOS ONE

Dear Dr. Gebeyew,

Thank you for submitting your manuscript to PLOS ONE. After careful consideration, we feel that it has merit but does not fully meet PLOS ONE’s publication criteria as it currently stands. Therefore, we invite you to submit a revised version of the manuscript that addresses the points raised during the review process.

We look forward to receiving your revised manuscript.

Kind regards,

Yibeltal Alemu Bekele, MpH

Academic Editor

PLOS ONE

Journal Requirements:

Funding was obtained from Debre Markos University. This work would not be possible without the financial support of the Research and Technology Transfer Directorate (RTTD) under grant number 177/01/17.

3. Please ensure that you refer to Figure 1, 2, 3, 4, and 5 in your text as, if accepted, production will need this reference to link the reader to the figure."

4. We note that Figure 2 in your submission contain [map/satellite] images which may be copyrighted. All PLOS content is published under the Creative Commons Attribution License (CC BY 4.0), which means that the manuscript, images, and Supporting Information files will be freely available online, and any third party is permitted to access, download, copy, distribute, and use these materials in any way, even commercially, with proper attribution. For these reasons, we cannot publish previously copyrighted maps or satellite images created using proprietary data, such as Google software (Google Maps, Street View, and Earth). For more information, see our copyright guidelines: http://journals.plos.org/plosone/s/licenses-and-copyright.

5. We note you have included a table to which you do not refer in the text of your manuscript. Please ensure that you refer to Table 5 and 6 in your text; if accepted, production will need this reference to link the reader to the Table.

Reviewers' comments:

Reviewer's Responses to Questions

**Comments to the Author**

1. Is the manuscript technically sound, and do the data support the conclusions?

Reviewer #1: Yes

Reviewer #2: Yes

2. Has the statistical analysis been performed appropriately and rigorously? 

Reviewer #1: Yes

Reviewer #2: Yes

3. Have the authors made all data underlying the findings in their manuscript fully available?

Reviewer #1: Yes

Reviewer #2: Yes

4. Is the manuscript presented in an intelligible fashion and written in standard English?

Reviewer #1: Yes

Reviewer #2: Yes

5. Review Comments to the Author

Reviewer #1: In the Title, change "Professional’s" to "Professionals’" for grammatical accuracy. In the Abstract, the background section clearly explains the significance and knowledge gap, but the statement "health system has a unified digital health center" could be rephrased for clarity. The Methods section adequately describes the study design and analysis approach but could briefly explain why SEM was specifically chosen. In the Results, key findings are presented clearly with relevant statistics, but it would be helpful to specify the strength of influence for perceived usefulness (PU), perceived trust (PT), and attitude. The Conclusion is well-linked to the findings but could suggest practical interventions. In the Introduction, the background provides a comprehensive overview of digital health data hubs, global data trends, and the Ethiopian context. However, the first paragraph is too broad, and the historical context on data storage feels unnecessary. The flow of the introduction is somewhat disjointed, and better transitions are needed between global trends, the local context, and the study's relevance. Additionally, there is repetition of concepts, especially regarding perceived ease of use, which could be consolidated. In the Methods, the cross-sectional design is well-suited for the research objectives, offering a snapshot of healthcare professionals' intentions at a specific point in time. The sample size calculation is well-explained, using a robust method for structural equation modeling (SEM) with sufficient statistical power. The use of stratified sampling ensures that diverse healthcare facilities (specialized, general, primary hospitals) are represented. The pre-test and detailed quality assurance measures, such as training and supervisor oversight, demonstrate a thorough approach to ensuring data validity and consistency. SEM is a suitable choice for analyzing the complex relationships between multiple constructs and latent variables.

Reviewer #2: Overall, this manuscript presents valuable insights into healthcare professionals' intentions regarding digital health data hubs in Ethiopia. The topic of digital health data hubs is highly relevant, especially in the context of achieving Sustainable Development Goals (SDGs). Your focus on Ethiopia adds an important dimension to the global conversation about digital health. These merits contribute to making your manuscript a valuable addition to the literature on digital health adoption among healthcare professionals, particularly in developing countries like Ethiopia.

General Feedback

1. Clarity and Conciseness:

a. The abstract is informative but could be more concise. Aim to summarize key findings in fewer words while maintaining clarity

b. Some sentences are complex, consider breaking them into shorter sentences for better readability.

Example from the introduction

Original: "Gradually, more and more people are using mobile devices and the Internet and intend to use them for healthcare services, as digital health (DH) has significantly accelerated the achievement of the Sustainable Development Goals (SDGs) and strengthened healthcare systems in Africa."

Suggested Revision: "More people are gradually using mobile devices and the Internet for healthcare services. Digital health (DH) has significantly accelerated progress toward achieving the Sustainable Development Goals (SDGs) and strengthening healthcare systems in Africa."

Example from Background Section:

Original: "To date, it is of great importance to examine the emerging cloud-based adoption at the organizational level Healthcare organizations have continually recorded data over time for customers, suppliers, and stakeholders to analyze the data and derive insights."

Suggested Revision: "It is important to examine emerging cloud-based adoption at the organizational level. Healthcare organizations have continually recorded data over time for customers, suppliers, and stakeholders to analyze this information and derive insights."

2. Structure:

a. Ensure that each section flows logically into the next. For example, the transition from the introduction to methods could be smoother by summarizing how the background leads to your research questions or hypotheses.

b. Consider using subheadings within sections (e.g., "Methods," "Results") to improve navigation through the document.

3. Methodology:

a. It might be helpful to provide more detail about how you ensured the validity and reliability of your survey instruments beyond mentioning pre-testing and PCA.

b. Clarify how you addressed potential biases in sampling or data collection.

4. Results Presentation:

a. Present results in a clear manner using tables or figures where appropriate (e.g., showing demographic data).

5. Discussion:

a. The discussion should connect back to your research questions/hypotheses more explicitly.

b. Consider discussing limitations earlier in this section rather than at the end; this can help contextualize your findings.

6. Conclusion:

Your conclusion summarizes findings well but could benefit from a stronger emphasis on implications for practice and future research directions.

Specific Feedback

Abstract:

a. The phrase "the health system has a unified digital health center" might need clarification - does it mean there is one central hub for all data?

b. Instead of stating "this study aims," use past tense since this is a completed study. "This study assessed..."

Introduction

a. The introduction provides good context but could be streamlined by focusing on key points relevant to your study's objectives.

b. Avoid excessive citations in introductory paragraphs; instead, synthesize information from multiple sources into cohesive statements.

Methods

a. Specify what type of healthcare professionals were surveyed (e.g., doctors, nurses) early on for context.

b. In describing SEM analysis, briefly explain why this method was chosen over others.

Results

a. Include descriptive statistics before diving into inferential statistics to give readers context about your sample.

b. Be cautious with terms like “significant” without specifying p-values initially; clarify what constitutes significance based on your analysis plan.

Discussion

a. Expand on how these findings relate to existing literature—what do they add or challenge?

b. Discuss practical applications of your findings more thoroughly—how can stakeholders implement changes based on this research?

Congratulations to the team on this great achievement.

6. PLOS authors have the option to publish the peer review history of their article (what does this mean? ). If published, this will include your full peer review and any attached files.

**Do you want your identity to be public for this peer review?** For information about this choice, including consent withdrawal, please see our Privacy Policy .

Reviewer #1: No

Reviewer #2: No

---

## [Author Response · Author response to Decision Letter 1]

1 Mar 2025

We thank you so much for giving constructive comments and suggestions for the manuscript paper. We have discussed these queries in detail below.

Part 1: Editors comments:

Question 1: Please ensure that your manuscript meets PLOS ONE's style requirements, including those for file naming.

Answer 1: Concerning Journal Requirements, we have covered all suggestions given for us and filled full the requirements of PLOS ONE's style requirements.

Question 2: Please state what role the funders took in the study. If the funders had no role, please state: "The funders had no role in study design, data collection and analysis, decision to publish, or preparation of the manuscript."

Answer 2: The funder had no role in study design, data collection and analysis, decision to publish, or preparation of the manuscript.

Question 3: Please ensure that you refer to Figures 1, 2, 3, 4, and 5 in your text as, if accepted, production will need this reference to link the reader to the figure."

Answer 3: Figures are referred to their appropriate place.

Question 4: we noticed that Figure 2 in your submission contains [map/satellite] images which may be copyrighted.

Answer 4: Based on the suggestion, the figure is removed from the submission.

Question 5. We note you have included a table to which you do not refer in the text of your manuscript. Please ensure that you refer to Tables 5 and 6 in your text; if accepted, production will need this reference to link the reader to the Table.

Answer 5: The tables have been cited in the appropriate place.

Question 6: Please include captions for your Supporting Information files at the end of your manuscript, and update any in-text citations to match accordingly.

Answer 6: We have put the caption of supporting information files at the end of the manuscript.

Question 7: Please review your reference list to ensure that it is complete and correct.

Answer 7: We have checked all references used in the manuscript

Part 2: Reviewers' comments

Response to Reviewer #1

C1: In the Title, change " Professional’s " to " Professionals’" for grammatical accuracy.

A1: The word has changed from "Professional’s" to "Professionals’"

C2: In the Abstract, the background section clearly explains the significance and knowledge gap, but the statement "health system has a unified digital health center" could be rephrased for clarity.

A2: the word is rephrased as “the health system has been coming to one central hub for all data”

C3: The Methods section adequately describes the study design and analysis approach but could briefly explain why SEM was specifically chosen.

A3: It is mentioned in this way. "Structural equation modeling (SEM) was used for the analysis. Because it is a more powerful multivariate technique to test and evaluate multivariate causal relationships." For more, however, structural equation modeling (SEM) is a little bit similar to regression analysis such as linear or logistic regression analysis, it is a powerful, multivariate technique in scientific investigations to test and evaluate multivariate causal relationships. Hence, it helps for examining linear causal relationships among variables, while simultaneously accounting for measurement error.

C4: In the Results, key findings are presented clearly with relevant statistics, but it would be helpful to specify the strength of influence for perceived usefulness (PU), perceived trust (PT), and attitude.

A4: The strength of influence of the significant variables such as PU, PT, and attitude are mentioned like this “perceived usefulness (PU: β = 0.576, p = 0.000), perceived trust (PT: β = 0.116, p = 0.022), and attitude (β = 0.143, p = 0.043) significantly and positively influenced health professionals’ intention to use digital health data hubs”.

C5: The Conclusion is well-linked to the findings but could suggest practical interventions.

A5: The conclusion is updated as given below.

Overall, the findings showed that 42.31% of health professionals have low intention to use digital health data hubs. These shall be needed to improve their intentions to use digital health data hubs through targeted interventions. Therefore, focusing on critical factors, such as perceived usefulness, trust, and attitude are crucial factors to reinforce their intention to use the system. Additionally, overcoming implementation challenges and building trust is critical to the successful integration and use of digital health data hubs.

C6: In the Introduction, the background provides a comprehensive overview of digital health data hubs, global data trends, and the Ethiopian context. However, the first paragraph is too broad, and the historical context on data storage feels unnecessary. The flow of the introduction is somewhat disjointed, and better transitions are needed between global trends, the local context, and the study's relevance. Additionally, there is the repetition of concepts, especially regarding perceived ease of use, which could be consolidated.

A6: The first paragraph has been modified and the flow of ideas has been adjusted. The repetition of concepts is updated.

C7: In the Methods, the cross-sectional design is well-suited for the research objectives, offering a snapshot of healthcare professionals' intentions at a specific point in time. The sample size calculation is well-explained, using a robust method for structural equation modeling (SEM) with sufficient statistical power. The use of stratified sampling ensures that diverse healthcare facilities (specialized, general, and primary hospitals) are represented. The pre-test and detailed quality assurance measures, such as training and supervisor oversight, demonstrate a thorough approach to ensuring data validity and consistency. SEM is a suitable choice for analyzing the complex relationships between multiple constructs and latent variables.

Regarding perceived ease of use, previous kinds of literature showed that perceived ease of use was a determinant factor for perceived usefulness, attitude, and intention to use. Hence, to explain this association, the word perceived ease of use was used again and again.

A7: Thank you so much for detailed comments.

Reviewer #2:

Overall, this manuscript presents valuable insights into healthcare professionals' intentions regarding digital health data hubs in Ethiopia. The topic of digital health data hubs is highly relevant, especially in the context of achieving Sustainable Development Goals (SDGs). Your focus on Ethiopia adds an important dimension to the global conversation about digital health. These merits contribute to making your manuscript a valuable addition to the literature on digital health adoption among healthcare professionals, particularly in developing countries like Ethiopia.

General Feedback

1. Clarity and Conciseness:

a. The abstract is informative but could be more concise. Aim to summarize key findings in fewer words while maintaining clarity

A1: The findings are summarized

b. Some sentences are complex, consider breaking them into shorter sentences for better readability.

Example from the introduction

Original: "Gradually, more and more people are using mobile devices and the Internet and intend to use them for healthcare services, as digital health (DH) has significantly accelerated the achievement of the Sustainable Development Goals (SDGs) and strengthened healthcare systems in Africa."

Suggested Revision: "More people are gradually using mobile devices and the Internet for healthcare services. Digital health (DH) has significantly accelerated progress toward achieving the Sustainable Development Goals (SDGs) and strengthening healthcare systems in Africa."

Example from Background Section:

Original: "To date, it is of great importance to examine the emerging cloud-based adoption at the organizational level Healthcare organizations have continually recorded data over time for customers, suppliers, and stakeholders to analyze the data and derive insights."

Suggested Revision: "It is important to examine emerging cloud-based adoption at the organizational level. Healthcare organizations have continually recorded data over time for customers, suppliers, and stakeholders to analyze this information and derive insights."

A2: Thank you so much for the surgical view, such comments are taken into consideration.

2. Structure:

a. Ensure that each section flows logically into the next. For example, the transition from the introduction to methods could be smoother by summarizing how the background leads to your research questions or hypotheses.

A3: Corrections were made for the flow of sections to logically connect the ideas.

b. Consider using subheadings within sections (e.g., "Methods," "Results") to improve navigation through the document.

A4: Heading and Subheadings are used for each section in detail as we see in the figure below.

3. Methodology:

a. It might be helpful to provide more detail about how you ensured the validity and reliability of your survey instruments beyond mentioning pre-testing and PCA.

A5: To ensure that the survey instruments are psychometrically sound, we have used a standard instrument and statistical methods. As we have mentioned in the data collection procedure and quality assurance section, The survey instrument has been adapted and modified from previous research. To make reliable and valid the instrument, the first task was, we assign a research team from different health departments who have profound knowledge and were trained in digital health to review and identify discrepancies regarding the research title. Then, they dug various literature on health professionals' intentions to use digital health data hubs. After that, the questionnaire was prepared. Before working on the actual survey, to check its validity, or quality of the survey, we conducted research consultations and pretests to check the measurement. Based on suggestions from medical experts including health professionals and research consultants, the questions were carefully and precisely worded. The pretest was also administered to check the reliability and item clarity. Based on their suggestions, the contents were modified, and the reliability of items greater than 70% was included in the final study. Then, during analysis, the completeness of the survey was checked, and other such as composite reliability, confirmatory factor analysis, and multivariate normality were measured, as mentioned in the method and result sections.

b. Clarify how you addressed potential biases in sampling or data collection.

A6: In order to control potential bias, the sampling method was calculated scientifically using an a priori structural equation modeling sample technique. The sample was enough participants to produce valid and generalizable results. The study participants were selected using systematic sampling methods, which helps to reduce selection bias. Furthermore, to ensure the reliability of the study, the first training was given for data collectors, supervisors, and researchers to follow the same steps in the same way for each measurement. This reduces bias regarding data collectors and multiple researchers. In addition, during data collection, they kept the circumstances as consistent as possible to reduce the influence of external factors that might create variation in the results.

4. Results Presentation:

a. Present results in a clear manner using tables or figures where appropriate (e.g., showing demographic data).

A7: For more understanding, tables and figures have been used to present the result. For example, the demographic data of the study is presented in Table 1.

5. Discussion:

a. The discussion should connect back to your research questions/hypotheses more explicitly.

A8: The research questions are discussed in detail discussion part.

b. Consider discussing limitations earlier in this section rather than at the end; this can help contextualize your findings.

A9: It is adjusted.

6. Conclusion:

Your conclusion summarizes findings well but could benefit from a stronger emphasis on implications for practice and future research directions.

A10: it is updated based on the suggestions.

Specific Feedback

Abstract:

a. The phrase "the health system has a unified digital health center" might need clarification - does it mean there is one central hub for all data?

A1: Yes

b. Instead of stating "this study aims," use past tense since this is a completed study. "This study assessed..."

A2: Well, it is corrected.

Introduction

a. The introduction provides good context but could be streamlined by focusing on key points relevant to your study's objectives.

A3: The background section has been updated well.

b. Avoid excessive citations in introductory paragraphs; instead, synthesize information from multiple sources into cohesive statements.

4: The citation has been updated well.

Methods

a. Specify what type of healthcare professionals were surveyed (e.g., doctors, nurses) early on for context.

A4: In that study area, there is one specialized referral hospital, one general hospital, and nine primary hospitals are used for referral by providing laboratory, pharmacy, radiology, intensive patient care, ophthalmic, psychiatry, obstetrics, and pediatric services. Health professionals working in these healthcare settings and who have experience in digital applications were included in the study. These health professionals are medical doctors, pharmacy, Anastasia, optometry, laboratory, radiology, health informatics, nurses, midwifery, psychiatry, and health officers.

b. In describing SEM analysis, briefly explain why this method was chosen over others.

A5: Structural equation modeling (SEM) is a powerful, multivariate technique in scientific investigations to test and evaluate multivariate causal relationships rather than regression analysis such as linear, and logistic analysis. Hence, we used SEM to examine causal relationships among variables, while simultaneously accounting for measurement error.

Results

a. Include descriptive statistics before diving into inferential statistics to give readers context about your sample.

A6: the sample of the survey is noted in the sociodemographic section.

b. Be cautious with terms like "significant" without specifying p-values initially; clarify what constitutes significance based on your analysis plan.

A7: Based on the review of previous research and according to the statistical rule, p< 0.05 is significant, hence, we used this cut-off point to say significant or not[1].

Discussion

a. Expand on how these findings relate to existing literature—what do they add or challenge?

A8: The discussion is updated according to your suggestion

b. Discuss practical applications of your findings more thoroughly—how can stakeholders implement changes based on this research?

A9: It is discussed in detail with practical application of the findings in the discussion section.

Finally, we would like to give great thanks to the Editors and Reviewers for your detailed comments, and suggestions by giving your golden time. If you any questions have, feel free to inform me.

[1] S. Greenland et al., "Statistical tests, P values, confidence intervals, and power: a guide to misinterpretations," European Journal of Epidemiology, vol. 31, no. 4, pp. 337-350, 2016/04/01 2016, doi: 10.1007/s10654-016-0149-3.

---

## [Decision Letter · Decision Letter 1]

24 Mar 2025

Health Care Professionals Intention to Use Digital Health Data Hub Working in East Gojjam Hospitals, Northwest Ethiopia: Technology Acceptance Modeling

PONE-D-25-02270R1

Dear Dr. Ayenew Sisay Gebeyew,

We’re pleased to inform you that your manuscript has been judged scientifically suitable for publication and will be formally accepted for publication once it meets all outstanding technical requirements.

Kind regards,

Yibeltal Alemu Bekele, MpH

Academic Editor

PLOS ONE

Reviewers' comments:

Reviewer's Responses to Questions

**Comments to the Author**

1. If the authors have adequately addressed your comments raised in a previous round of review and you feel that this manuscript is now acceptable for publication, you may indicate that here to bypass the “Comments to the Author” section, enter your conflict of interest statement in the “Confidential to Editor” section, and submit your "Accept" recommendation.

Reviewer #2: All comments have been addressed

2. Is the manuscript technically sound, and do the data support the conclusions?

Reviewer #2: Yes

3. Has the statistical analysis been performed appropriately and rigorously? 

Reviewer #2: Yes

4. Have the authors made all data underlying the findings in their manuscript fully available?

Reviewer #2: Yes

5. Is the manuscript presented in an intelligible fashion and written in standard English?

Reviewer #2: Yes

6. Review Comments to the Author

Reviewer #2: In the first round of the review, I provide extensive comments on the submitted manuscripts. Following the comments provided, the authors have gone ahead to review, update, and resubmit the updated version of their manuscript. After reviewing the latest version, the manuscript can now be accepted.

7. PLOS authors have the option to publish the peer review history of their article (what does this mean? ). If published, this will include your full peer review and any attached files.

**Do you want your identity to be public for this peer review?** For information about this choice, including consent withdrawal, please see our Privacy Policy .

Reviewer #2: No

---

## [Editor Report · Acceptance letter]

PONE-D-25-02270R1

PLOS ONE

Dear Dr. Gebeyew,

I'm pleased to inform you that your manuscript has been deemed suitable for publication in PLOS ONE. Congratulations! Your manuscript is now being handed over to our production team.

Kind regards,

on behalf of

Mr. Yibeltal Alemu Bekele

Academic Editor

PLOS ONE